# Anthropogenic amplification of biogenic secondary organic aerosol production

Yiqi Zheng[1,2,*], Larry W. Horowitz[3], Raymond Menzel[3], David J Paynter[3], Vaishali Naik[3], Jingyi Li[4], Jingqiu Mao[1,2,*]

[1]Geophysical Institute, University of Alaska Fairbanks, Fairbanks, AK, USA
[2]Department of Chemistry and Biochemistry, University of Alaska Fairbanks, AK, USA
[3]NOAA Geophysical Fluid Dynamics Laboratory, Princeton, NJ, USA
[4]School of Environmental Science and Engineering, Nanjing University of Information Science and Technology, Nanjing, China

*Correspondence to Yiqi Zheng (zhengyiqi1989@gmail.com) and Jingqiu Mao (jmao2@alaska.edu)

## Abstract

Biogenic secondary organic aerosols (SOA) contribute to a large fraction of fine aerosols globally, impacting air quality and climate. The formation of biogenic SOA depends on not only emissions of biogenic volatile organic compounds (BVOCs) but also anthropogenic pollutants including primary organic aerosol, sulfur dioxide ($SO_2$), and nitrogen oxides ($NO_x$). However, the anthropogenic impact on biogenic SOA production (AIBS) remains unclear. Here we use the decadal trend and variability of observed OA in the southeast US, combined with a global chemistry-climate model, to better constrain AIBS. We show that the reduction in $SO_2$ emissions can only explain 40% of the decreasing decadal trend of OA in this region, constrained by the low summertime month-to-month variability of surface OA. We hypothesize that the rest of OA decreasing trend is largely due to reduction in $NO_x$ emissions. By implementing a scheme for monoterpene SOA with enhanced sensitivity to $NO_x$, our model can reproduce the decadal trend and variability of OA in this region. Extending to centennial scale, our model shows that global SOA production increases by 36% despite BVOC reductions from preindustrial period to present day, largely amplified by AIBS. Our work suggests a strong coupling between anthropogenic and biogenic emissions in biogenic SOA production that is missing from current climate models.

## 1. Introduction

Terrestrial vegetation emits more than 1 Pg per year of BVOCs (Guenther et al., 2012), leading to a major source of SOA in the atmosphere (Goldstein and Galbally, 2007). SOA exerts significant impacts on climate, air quality and human welfare (Shrivastava et al., 2017; Pye et al., 2021), but is not well represented in climate models. Global climate models differ largely in simulated SOA burden, variability, and radiative effects (Tsigaridis et al., 2014) due to complexity associated with emission of precursors, multiphase chemical and physical processes, aging, radiative properties, and other processes (Shrivastava et al., 2017). Many climate models simply scale SOA yield with BVOC precursors (Horowitz et al., 2020; Carslaw et al., 2013; Koch et al., 2011).

Current understanding of biogenic SOA formation has advanced far beyond this simple scaling of BVOC emissions. SOA formation from BVOC oxidation is largely dependent on its oxidants ($OH/O_3/NO_3$) and the yields show non-linear behavior under different $NO_x$ conditions (Ng et al., 2017; Presto et al., 2005). One advanced scheme is the Volatility Basis Set (VBS), in which intermediate semivolatile products from the oxidation of BVOCs are grouped into volatility bins and can reversibly condense onto pre-existing organic aerosols (Donahue et al., 2006; Pye et al., 2010). VBS accounts for the dependence of SOA formation on atmospheric oxidants, $NO_x$-dependent chemical regimes, POA and temperature. Some studies showed that VBS schemes underestimated observations and that photochemical aging schemes with varying complexity may improve simulation results in different regions and seasons (Zheng et al., 2015; Robinson et al., 2007; Oak et al., 2022). Another pathway is through reactive uptake of smaller molecules onto aqueous aerosols. Several isoprene oxidation products, such as epoxides (IEPOX) (Paulot et al., 2009) and glyoxal (Liggio, 2005; Li et al., 2016), though often not directly condensable due to their high equilibrium vapor pressure, can undergo aqueous phase reactions and oligomerize in the condensed phase. The detailed mechanism is complicated by aerosol acidity, composition, and coating (Shrivastava et al., 2017). These advancements highlight the role of anthropogenic emissions modulating biogenic SOA formation through nitrogen oxides ($NO_x$), $SO_2$ and primary organic aerosol (POA).

One major uncertainty is to what extent anthropogenic emissions modulate biogenic SOA formation. In the southeast US (SEUS), a region largely covered by natural vegetation and also heavily populated, organic aerosol shows a decreasing trend in the recent two decades (Kim et al., 2015; Attwood et al., 2014), likely due to reductions in POA (Blanchard et al., 2016; Ridley et al., 2018; Liu et al., 2023), sulfate and aerosol water (Christiansen et al., 2020; Ridley et al., 2018; Marais et al., 2017; Malm et al., 2017; Blanchard et al., 2016; Liu et al., 2023) and $NO_x$ (Zheng et al., 2015; Xu et al., 2015; Pye et al., 2019). Several studies suggest that $SO_2$ largely modulates SOA through reactive uptake of IEPOX (Pye et al., 2013; Marais et al., 2017), but the acidity-catalyzed sulfate uptake mechanism appears to overestimate the trend of OA reduction in the SEUS (Zheng et al., 2020). The role of $NO_x$ remains unclear. While SOA yield generally decreases with $NO_x$ level due to fragmentation of large molecules (Kroll and Seinfeld, 2008), recent studies show that $NO_x$ can in fact increase SOA production (Zheng et al., 2015; Xu et al., 2015; Pye et al., 2019; Pullinen et al., 2020). The combined effect of $NO_x$, $SO_2$ and POA can be significant (Carlton et al., 2010; Hoyle et al., 2011), but remain unconstrained by ambient observations.

Here we use the decadal trend and variability of observed OA in the southeast US, combined with other observational datasets and a global chemistry-climate model (GFDL AM4.1) (Horowitz et

al., 2020), to better constrain the anthropogenic impact on biogenic SOA production (AIBS). We use three schemes (summarized in Table 1 and detailed in Methods) to investigate the AIBS from decadal to centennial time scales.

## 2. Methods
### 2.1 GFDL AM4.1
The Geophysical Fluid Dynamics Laboratory (GFDL)'s Atmospheric Model version 4.1 (AM4.1) (Horowitz et al., 2020) is a three-dimensional global chemistry-climate model that includes interactive simulation of stratospheric chemistry and tropospheric $O_3$-$NO_x$-CO-VOC and bulk aerosol chemistry, allowing explicit treatment of aerosol reactive uptake of IEPOX and glyoxal (Li et al., 2016, 2018; Mao et al., 2013). AM4.1 has 49 vertical levels from surface to 1Pa (~80km). We conduct AM4.1 simulations at a horizontal resolution of 1°×1.25° latitude by longitude and a main dynamical atmosphere time step of 30 minutes. Annual varying historical anthropogenic emissions from pre-industrial era to present day (1849 to 2016) are from the Community Emissions Data System (CEDS) (Hoesly et al., 2018) and the data set of van Marle et al. (2017), which are developed in support of the Coupled Model Intercomparison Project Phase 6 (CMIP6). Global fire emissions are based on Global Fire Emissions Database version 4 (GFED4), the Fire Modeling Intercomparison Project (FireMIP), visibility-observations and Global Charcoal Database (GCD). Biogenic isoprene and monoterpene emissions are calculated online by the Model of Emissions of Gases and Aerosols from Nature version 2.1 (MEGAN2.1), using empirical functions of plant-functional-type (PFT)-specific emission basal factors, leaf area index (LAI), temperature and light. Dependence of soil moisture, $O_3$ and $CO_2$ are neglected due to large uncertainties. LAI values follow an annual cycle of the year 1992 and PFTs are prescribed at the 1992 level. The gas-phase and aerosol chemistry is detailed in Horowitz et al. (2020), in which heterogenous reactive uptake of $HO_2$, $HO_2$, $NO_2$, $N_2O_5$, $NO_3$, $SO_2$, IEPOX and glyoxal onto aerosol surfaces are included. Dry and wet deposition of gases are described in Paulot et al. (2016). More details could be found in Horowitz et al. (2020) and Dunne et al. (2020). Radiative effects of SOA is calculated assuming SOA is externally mixed from other aerosols (Horowitz et al., 2020), although ISOA is formed through sulfate uptake in the chemistry module.

We perform simulations for years 1998-2016 for present day (PD) and 1870-1888 for pre-industrial period (PI). In each simulation, the first two years are discarded as spin-up. The remaining 17 years are used for analysis. The PD simulations are nudged with reanalysis winds from NCEP-DOE Reanalysis 2. The PI simulations are free running with no nudging. All simulations are driven by observed or reconstructed sea surface temperature and sea-ice (Horowitz et al., 2020). In the two PI simulations, we scale up the isoprene and monoterpene emission basal factors by 35% to account for the higher natural vegetation cover at pre-industrial period than today, equivalent to a 26% reduction of natural vegetation cover from PI to PD (Unger, 2014). We apply this single scaling factor to BVOC emission basal factors as an idealized study instead of using reconstructed land cover type and LAI to avoid uncertainties in historical vegetation reconstructions.

### 2.2 Modeling of SOA formation
In GFDL AM4.1, SOA is composed of anthropogenic SOA (ASOA), isoprene SOA (ISOA) and monoterpene SOA (TSOA). ASOA is formed through the oxidation of $C_4H_{10}$ by OH in all simulations. In the default "Simple" scheme, ISOA and TSOA are assumed to be produced with a

pseudo-emission equivalent to a 10% per-carbon yield of the interactively calculated isoprene and monoterpene emissions, respectively. This 10% yield in the Simple scheme is consistent with previous model versions GFDL AM3 and AM4.0 and within the range of estimates suggested by other studies. For example, a chemical transport model GEOS-Chem assumed a 3% yield for isoprene and a 10% yield of monoterpene emissions (Pye et al., 2010; Pai et al., 2020). However, a study using a more complex scheme suggested a SOA yield from isoprene of 13% per carbon (Bates and Jacob, 2019).

We implement a complex (CMPX) SOA scheme in GFDL AM4.1, in which isoprene and monoterpenes are oxidized by OH, $O_3$ and $NO_3$ to form ISOA and TSOA. ISOA is computed through the aqueous-phase uptake of IEPOX and glyoxal onto sulfate aerosol. The uptake coefficients for IEPOX and glyoxal are set to 0.001, different than previous studies using higher or acidity-dependent uptake coefficients (Marais et al., 2016; Lin et al., 2014a). This is supported by the OA month-to-month variability (MMV) in summer and its decadal trend over SEUS, as a previous model with acidity-dependent uptake coefficients shows too high of MMV and too much OA in the early 2000s (Zheng et al., 2020). The uptake rate coefficients can be even lower due to the effect of aerosol-phase state (Zhang et al., 2018b). To avoid uncertainties associated with aerosol acidity, relative humidity, and coating effect, we here apply uptake coefficient of 0.001 for both IEPOX and glyoxal. This leads to good agreement between our model and observation in SEUS on both OA magnitude and summertime MMV (Figure1, S1, and S2).

In the updated "CMPX" scheme, TSOA is calculated by a 4-product Volatility Basis Set (VBS) summarized in Table 1. Organic peroxy radicals ($RO_2$) formed from OH- and $O_3$-initiated oxidation of monoterpene can react with NO under high-$NO_x$ conditions and with $HO_2$ under low-$NO_x$ conditions. The low-$NO_x$ pathway ($RO_2$+$HO_2$) has higher yields for SOA than the high-$NO_x$ pathway ($RO_2$+NO) (Pye et al., 2010; Zheng et al., 2015). The branching ratio between the low- versus high-$NO_x$ pathways are defined as:

$$\beta_{NO} = \frac{k_{RO2+NO} * [NO]}{k_{RO2+NO} * [NO] + k_{RO2+HO2} * [HO_2]}$$

Where $k_{RO2+NO}$ and $k_{RO2+HO2}$ represent the reaction rate coefficients of $RO_2$+NO and $RO_2$+$HO_2$, respectively. At nighttime, the $NO_3$-initiated oxidation of monoterpenes has a high yield of organic nitrates and contributes a significant amount of SOA (Ng et al., 2017). The surrogate TSOA products are implemented in addition to the original gas-phase monoterpene oxidation chemistry in AM4.1, and the implementation does not doubt count reductions of OH, $O_3$ and $NO_3$. There is little difference in the concentration of these gases between the CMPX and Simple simulations. The gas-phase chemistry has been validated in Horowitz et al. (2020) and in Figure S3 in which we show that summertime surface $O_3$ and $NO_2$ in SEUS well reproduce their observed decreasing trend.

Such semi-empirical partitioning-based VBS schemes have been widely used in chemistry-climate models and Earth system models (e.g. Zheng et al., 2015; Tilmes et al., 2019). Recent research show that these schemes may underestimate SOA formation without considering further aging processes, such as oligomerization in the organic phase and aqueous-phase reactions (Hu et al., 2013; Yu et al., 2021; Oak et al., 2022 and references therein). One major recent identified

explicit mechanism is the formation of monoterpene-derived highly oxygenated molecules (HOMs) through the autooxidation of peroxy radicals (Crounse et al., 2013; Ehn et al., 2014; Pye et al., 2019). Mechanistic schemes of monoterpene-derived SOA have been developed with varying complexity at a cost of more tracers and reactions (Pye et al., 2019; Berkemeier et al., 2020; Pullinen et al., 2020; Yu et al., 2021), which may not be mature for a global climate model as part of an Earth system model considering large uncertainties associated with multi-phase processes and increased computational cost. In this study, in addition to the semi-empirical VBS scheme, we implement a simplified photochemical aging parameterization to the semivolatile oxidation products of terpenes in the CMPX scheme (CMPX_ag) (Zheng et al., 2015), to account for the decrease in volatility as a result of OH oxidation (Donahue et al., 2012). We apply a rate constant of $k_{OH} = 4\times10^{-11}$ cm$^3$ molec$^{-1}$ s$^{-1}$ (Robinson et al., 2007), in line with recent estimates of 2-4 $\times10^{-11}$ cm$^3$ molec$^{-1}$ s$^{-1}$ for terpene SOA (Donahue et al., 2012; Isaacman-VanWertz et al., 2018). Including the aging scheme in CMPX does not increase computational cost notably. This simplified aging scheme does not explicitly represent up-to-date knowledge of SOA chemistry but similarly increases the SOA burden as well as the sensitivity of SOA to NO$_x$, improving the model underestimate of SOA by the VBS scheme. The details of the Simple, CMPX, and CMPX_ag schemes are summarized in Table 1.

**2.3 Observational datasets**

For model evaluation we use long-term measurements of organic aerosol (OA) or organic carbon (OC). We do not use explicit SOA tracers in this study because it is not suitable to use short-term observations to validate simulated results from a chemistry-climate model like AM4.1, in which meteorology is not offline specified by reanalysis data but is free running in the dynamic core. However, long-term (covering at least months to years) measurement of explicit SOA species is not available. We use filter measurement of organic carbon from two surface aerosol measurement networks in the US: IMPROVE (the Interagency Monitoring of Protected Visual Environments) (Solomon et al., 2014) and SEARCH (the SouthEastern Aerosol Research and Characterization) (Edgerton et al., 2005). IMPROVE and SEARCH report daily average organic carbon measurements every 3 days. We focus on SEUS which is both heavily vegetated and populated. We select 20 IMPROVE sites and 3 SEARCH rural sites within the SEUS region (29-37°N, 74-96°W) and calculate monthly average of OA across these sites for each network (see site locations in Figure S4). We apply a seasonal-dependent ratio to convert organic carbon to OA mass: 2.2 in June-July-August, 1.8 in December-January-February and 1.9 in other months (Philip et al., 2014). In Section 3.1, we calculate the absolute trend of a variable as the slope of the regression line of the variable's value versus time, and we calculate the relative trend (represented by "m" in Figure 1) as the absolute trend divided by the variable's 2000-2016 average.

We also compare modeling results to OA measurement by Aerosol Chemical Speciation Monitor (ACSM). We select 3 European sites from the ACTRIS (the Aerosol, Clouds and Trace Gases Research Infrastructure) network (Crenn et al., 2015): Hyytiala (Finland), Puy de Dome (France) and Birkenes II (Norway); two sites from the ARM (Atmospheric Radiation Measurement) network (Uin et al., 2019): Southern Great Plains (US) and Manacapuru, Amazonia (Brazil). These sites are covered by natural vegetation and have more than a year's worth of data available. We average the original hourly OA measurement to monthly mean data for these sites to compare with modeling results.

## 3. Results

### 3.1 Decadal trend of summertime OA in SEUS and its variability

The SEUS is a region heavily influenced by both biogenic and anthropogenic emissions (Mao et al., 2018). In the last two decades, organic aerosol shows a decreasing trend, resulting from reductions in anthropogenic pollutants including $SO_2$ and $NO_x$ (Marais et al., 2017; Blanchard et al., 2016; Ridley et al., 2018). The CMPX and CMPX_ag scheme successfully reproduce the summertime surface OA concentrations measured from the IMPROVE and SEARCH networks at 4 and 5.5 $\mu g/m^3$, respectively (Figure 1a). The Simple scheme has a significant overestimate (~ 7 $\mu g/m^3$).

We first examine the simulated decadal OA trend in the SEUS against filter-based measurements from IMPROVE and SEARCH networks. From 2000 to 2016, the measured summer OA declines by -0.13 $\mu g/m^3$/year from SEARCH and by -0.09 $\mu g/m^3$/year from IMPROVE, both at a reduction rate of -2.3%/year (Figure 1a). This decreasing trend is well reproduced by the CMPX_ag simulation with a decrease of -0.11 $\mu g/m^3$ (-2.0%) per year, and a smaller decrease of -0.06 $\mu g/m^3$ (-1.4%) per year with the CMPX scheme. Considering the varying reduction trends among different sites (Figure S4), both the CMPX and CMPX_ag schemes well reproduce the SEUS OA trend in general. In contrast, the Simple scheme shows a slight increase (+0.7%/year) in surface OA due to lack of AIBS and little change of POA in 2000-2016 in this region (Figure 1c).

We further examine the summertime month-to-month variability of surface OA. We find that both CMPX_ag and CMPX schemes can well reproduce the low summertime month-to-month variability of surface OA (standard deviation smaller than 2 $\mu g/m^3$) constrained by IMPROVE and SEARCH measurements (Figure S2), using fixed uptake coefficients ($\gamma$=0.001) of IEPOX and glyoxal. This summertime month-to-month variability was found to be too high (standard deviation up to 5 $\mu g/m^3$) in the early 2000s using an acidity-dependent IEPOX reactive uptake scheme (Marais et al., 2016, 2017), pointing to additional species besides $SO_2$ driving the decreasing OA trend.

One unique feature of the CMPX_ag simulation is the dominance of TSOA (Figure 1), mainly through enhanced sensitivity of TSOA production to $NO_x$. Such dominance of TSOA in this region is also supported by recent field observations (Xu et al., 2018; Zhang et al., 2018a). We find TSOA contributes to 60% of the surface OA trend in the CMPX_ag scheme, mainly through $NO_x$ reduction. The $NO_3$-initiated pathway contributes to the majority of surface TSOA decrease (Figure S5), resulting from the rapid decrease of $NO_3$ (Figure 1d) (Boyd et al., 2017; Rollins et al., 2012). Compared to the CMPX scheme, the dominant contribution of TSOA is largely due to the OH aging effect, which amplifies the SOA yield from all monoterpene oxidation channels. As a result, we find that $NO_x$ reduction accounts for 60% of OA decrease in SEUS. This enhanced sensitivity to $NO_x$, resonates with recent developments on monoterpene-derived highly oxygenated organic molecules or autooxidation (Pye et al., 2019), highlighting the importance of $NO_x$ in AIBS.

ISOA contributes to 40% of surface OA trend in the CMPX_ag scheme, mainly through $SO_2$ reduction. The decrease in surface ISOA, at -0.05 $\mu g/m^3$/year, is associated with the strong reduction in sulfate (-7%/year). The rapidly decreasing sulfate, $NO_x$ and $O_3$ in the model are

consistent with observations over the SEUS (Figure S3) and previous studies (Zheng et al., 2020; Wells et al., 2021; Simon et al., 2015). In contrast to Marais et al. (2017), we find that this nondominant role of ISOA brings model into much better agreement with observations, especially on the low summertime month-to-month variability of surface OA (standard deviation smaller than 2 $\mu$g/m$^3$) constrained by IMPROVE and SEARCH measurements (Figure S2) (Zheng et al., 2020). The observed summertime month-to-month variability also implies a weaker dependence of OA to sulfate aerosols in this region than as shown in Marais et al. (2017), highlighting the importance of TSOA.

We find a similar trend of summer OA column concentration to the surface OA trend in the model. The CMPX_ag simulation suggests a decreasing trend in summer OA column concentration, driven by both TSOA (-0.13 mg/m$^2$/year) and ISOA (-0.12 mg/m$^2$/year) (Figure 1b). Similar to the surface, the aging effect increases the column production of TSOA in CMPX_ag and its sensitivity to changes in NO$_x$ compared with the CMPX scheme.

**3.2 Present-day OA in vegetated regions and global budget**
We further evaluate the modeled surface OA against measurements by Aerosol Chemical Speciation Monitor (ACSM) in other vegetated regions in the Amazon, Europe and US (Figure 2). In the Amazon region, the CMPX_ag scheme successfully reproduces the high surface OA concentration from August to November and low OA in other months (Figure 2c). The Simple scheme greatly overestimates surface OA in all seasons because of its high SOA yield (10%) from isoprene emissions. The CMPX scheme well reproduces the low OA concentrations from January to July but only predicts half of observed OA in months with high OA concentrations. In the 3 European sites from the ACTRIS network (Figure 2d~f), all model simulations underestimate measured OA. One possible reason is uncertainties associated with BVOC emissions and biogenic SOA. Jiang et al. (2019) showed that MEGAN overestimates isoprene emission but underestimates monoterpene emissions in Europe by a factor of 3. At the US Southern Great Plains site from the ARM network (Figure 2b), the CMPX_ag and CMPX schemes successfully capture the measured OA seasonal variation but underestimate OA magnitude. In the SEUS compared to filter measurements (Figure 2a), all simulations show lower OA in winter than observations, likely due to an underestimate of wintertime emissions of POA (Tsigaridis et al., 2014; Liu et al., 2021). In general, the updated CMPX_ag and CMPX schemes agree well with observations in the Amazon and US where biogenic emissions are high. The good performance of the CMPX_ag scheme in the Amazon, better than the CMPX scheme, gives us confidence that the traditional VBS in the CMPX scheme may underestimate the contribution of TSOA and its sensitivity to NO$_x$.

Globally, the SOA burden from the Simple, CMPX and CMPX_ag schemes are 0.99, 0.50 and 1.05 Tg, respectively, and their SOA production rates are 82, 40 and 69 Tg/year (Figure 3), in agreement with other global modeling studies. The AeroCom phase II model intercomparison summarizes a median SOA source of 51 Tg/year with a range between 16 to 121 Tg/year (Tsigaridis et al., 2014), although top-down methods indicate SOA source could be up to 50-380 Tg/year (Spracklen et al., 2011). Uncertainties associated with BVOC emissions contribute to the wide spread of SOA estimate by global models. In GFDL AM4.1, annual isoprene and monoterpene emissions are computed to be 505±14 and 137±5 Tg/year, respectively (Figure 3), in line with previous estimates (Guenther et al., 2012).

Detailed SOA budgets for the three schemes are summarized in Table 2. The CMPX and CMPX_ag schemes have much less ISOA than the Simple scheme as the latter has high pseudo emission of isoprene SOA, which is 10% in GFDL AM4.1 as compared to 3% used in other models like GEOS-Chem (Pai et al., 2020; Henze and Seinfeld, 2006). ISOA (22.2 Tg/year) and TSOA (14.4 Tg/year) in the CMPX scheme are consistent with previous estimate by GEOS-Chem (Pai et al., 2020; Zheng et al., 2020). The CMPX_ag scheme has higher TSOA (44 Tg/year) than CMPX and Simple due to the aging effect of semivolatile oxidation products from terpenes (Figure 3), and is close to the high end of estimate (12.7-40 Tg/year) by AeroComII (Tsigaridis et al., 2014). ASOA is often neglected by global models despite an estimate of 13.5 Tg/year suggesting ASOA as a non-negligible source (Tsigaridis et al., 2014). In GFDL AM4.1, ASOA (3.3 Tg/year) only considers oxidation of $C_4H_{10}$, which does not well represent all ASOA and warrants further research.

**3.3 Centennial change in biogenic SOA and direct radiative forcing**
We now extend our analysis of AIBS from the decadal scale to the centennial scale. To represent the higher natural vegetation cover during PI, we scale up isoprene and monoterpene emission basal factor in the PI simulations by 35%, equivalent to a 26% reduction of natural vegetation cover from PI to PD (Unger, 2014). This simple scaling should be considered as an idealized study to avoid uncertainties associated with historical vegetation reconstruction and the complex role of $CO_2$ including both fertilization and inhibition effects. From 1870s to 2000s, the simulated isoprene emissions decrease from 632±15 to 505±14 Tg/year (-20%) and monoterpene emissions decrease from 161±5 to 137±5 Tg/year (-15%) (Figure 3a, maps in Figure S6), consistent with previous studies (Heald and Spracklen, 2015).

Despite the reduction in BVOC emissions from PI to PD, we show a significant increase of biogenic SOA (Figure 3b, maps in Figure S7 and S8), resulting from increase in anthropogenic emissions and amplified by AIBS. With an increase by 1.4, 7, and 4 for emissions of POA, $SO_2$ and $NO_x$, total SOA production increases by 36% and its burden increases by 42% (in the CMPX_ag scheme). ASOA, ISOA and TSOA contribute 17%, 62%, and 21% to the changes in total SOA production, respectively. In contrast, the Simple scheme shows a decrease of SOA production following the reduction in BVOC emissions. The large increase of SOA from PI to PD differs from previous estimates (Spracklen et al., 2011; Heald and Spracklen, 2015; Zhu et al., 2019; Scott et al., 2017; Lin et al., 2014b; Heald and Geddes, 2016; Hoyle et al., 2009), largely due to AIBS constrained by observations.

The total PI-to-PD SOA rise is largely dominated by ISOA (62%), resulting from the strong increase in anthropogenic $SO_2$ emissions and uptake of IEPOX and glyoxal onto sulfate aerosols. The global burden of sulfate aerosol has doubled from 0.7 Tg at PI to 1.6 Tg at PD, with large increase over the tropics, SEUS, and Eurasia (Figure S9). The increase in TSOA is due to both increased $NO_x$ emissions and POA emissions. In contrast to the decadal trend where $\beta_{NO}$ barely changes, the PI-to-PD increase of TSOA due to the change of $NO_x$ is suppressed by the shift of $\beta_{NO}$. The branching ratio $\beta_{NO}$ increases from a global average of 0.32 at PI to 0.61 at PD (Figure S10), indicating a shift from low-$NO_x$ pathway (higher yields) to high-$NO_x$ pathway (lower yields) for the OH- and $O_3$-initiated oxidation. These competing effects lead to a net +10% change in TSOA production and a +14% increase in burden from PI to PD. The PI-to-PD change in TSOA in the CMPX scheme is small (-7% in production and +6% in burden). Increased POA provides

more organic mass for monoterpene oxidation products to condense on, especially in central Africa and central South America (Figure S9).

The large increase of biogenic SOA leads to a cooling direct radiative forcing (DRF) from PI to PD, opposed to the warming suggested by the Simple scheme. DRF is usually defined as the difference between PI and PD direct radiative fluxes at top-of-atmosphere under all-sky conditions. We show in Figure 4 the global instantaneous DRF at top-of-atmosphere of -(26–44) mW/m$^2$, comparable to that of POA (-98 mW/m$^2$). In contrast, the Simple scheme shows a warming DRF of +17 mW/m$^2$, largely due to lack of AIBS. The DRF of SOA in the updated schemes resides within reported AeroComII estimates, which ranges from -210 to -10 mW/m$^2$, with a mean value of -60 mW/m$^2$ and a median value of -20 mW/m$^2$ (Myhre et al., 2013). Due to this increase of SOA burden, our results may also imply a large indirect radiative forcing from biogenic SOA that is missing from previous work (Carslaw et al., 2013).

## 4. Summary

Our work suggests a strong coupling between anthropogenic and biogenic emissions in biogenic SOA production. Constrained by observations in SEUS, we show that the summertime OA decreasing trend is likely driven by reduction in both $NO_x$ and $SO_2$ emissions, through TSOA and ISOA. First, in a previous study (Zheng et al., 2020) we prove that the scheme of acidity-catalyzed aqueous ISOA formation (Marais et al., 2016) strongly overestimates summertime month-to-month variability of surface OA, therefore in this study we use fixed uptake coefficients for isoprene oxidation products to avoid uncertainties associated with acidity, relative humidity, and coating effect. Second, both the CMPX and CMPX_ag schemes reproduce the observed OA magnitude and the decadal trend in SEUS, in which $SO_2$ alone cannot explain this trend. The CMPX_ag scheme shows a faster OA decrease and better agrees with long-term filter measurement, which is largely driven by $NO_x$ (60%). Third, the CMPX_ag scheme successfully reproduces the observed OA magnitude and seasonal cycle in Amazon, outperforming the CMPX and Simple schemes. Our results point to the importance role of $NO_x$ on modulating biogenic SOA, in line with recent understanding on autooxidation (Crounse et al., 2013; Ehn et al., 2014; Pye et al., 2019), although further studies are warranted. For example, the CMPX_ag scheme with a simplified aging parameterization does not mechanistically represent the most up-to-date understanding of HOMs and organic nitrates (Takeuchi and Ng, 2019; Berkemeier et al., 2020; Pullinen et al., 2020; Yu et al., 2021). Long-term measurement of ISOA and TSOA tracers across different regions and seasons are needed to develop future mechanistic SOA schemes that are suit for global climate models with minimal computational cost. In this study, the success of the updated schemes in capturing the observed OA trend and month-to-month variability provides confidence in model simulations over longer time scales.

At centennial scale, atmospheric SOA mass increases significantly from PI to PD despite reductions in BVOC emissions, posing a top-of atmosphere instantaneous radiative forcing of -(26–44) mW/m$^2$. ISOA dominates the total SOA change as a result of a significant rise in global sulfate aerosol from PI to PD, especially in the fast-developing regions like Africa, Middle East, India, and China. POA increases greatly in central Africa and central South America as well as India and east China, which enhances TSOA production. The significant increase in SOA due to AIBS in these regions poses new challenges to meet the World Health Organization's recommendation on annual fine particulate matter exposure (5 μg/m$^3$) (Pai et al., 2022). Under

future scenarios with reduced emissions of $SO_2$, $NO_x$ and POA, the AIBS may indicate larger reductions in SOA than current model predictions, but their relative importance cannot be linearly extrapolated based on PI and PD simulations. Model simulations with future emission scenarios are needed, which is beyond the scope of this study.

The updated SOA scheme in GFDL AM4.1 shows an advance in representing vegetation-chemistry-climate interactions than the default model which assumes fixed yields of SOA from biogenic hydrocarbons, although a variety of uncertainties still exist in the evaluation of SOA and its climate impact. First, the model likely underestimates wintertime POA in the US, total OA in Europe and anthropogenic SOA globally. Second, the model does not consider absorbing SOA or brown carbon which could form from biomass burning and aging (Tsigaridis and Kanakidou, 2018). The model applies the same optical parameters for all SOA as hydrophilic POA. Third, other than the uncertainties discussed above about acidity-dependency, formation of HOMs and organic nitrates, other properties that influence the multiphase growth of SOA, including coating and viscosity, are also not implemented in our model (Shrivastava et al., 2017). Finally, the model does not consider nucleation of extremely low volatile compounds from BVOC oxidation, which may increase SOA in pristine environments in the pre-industrial period, thus reducing the PI-to-PD radiative forcing of SOA (Gordon et al., 2016; Zhu et al., 2019). These uncertainties warrant further research in studies on anthropogenic-influenced SOA in climate models.

**Acknowledgements**

We acknowledge funding from NOAA grant NA18OAR4310114 and NASA grant 80NSSC21K0428. We thank Fabien Paulot and Songmiao Fan for internal GFDL review and support from GFDL's Model Development Team, Modeling Systems Division, Operations group, and the RDHPCS supercomputing resources. We also acknowledge the Electric Power Research Institute (EPRI) and Southern Company for support of the SEARCH network and Atmospheric Research & Analysis, Inc; the US Environmental Protection Agency (EPA) for support of the IMPROVE network and Air Quality System; the European Union's Horizon 2020 research and innovation programme under grant agreement No 654109 for support of the ACTRIS network; and the Atmospheric Radiation Measurement (ARM) user facility, a U.S. Department of Energy (DOE) office of science user facility managed by the Biological and Environmental Research Program.

**Author Contributions**

Conceptualization: YZ, JM

Methodology: YZ, LWH, RM, DJP, VN, JM

Investigation: YZ, LWH, JM

Writing—original draft: YZ, JM

Writing—review & editing: YZ, LWH, RM, DJP, VN, JL, JM

**Data availability**

The IMPROVE filter OA and sulfate data is available at http://views.cira.colostate.edu/iwdw/. The ACTRIS ACSM OA data is available at https://actris.nilu.no/. The ARM ACSM OA data is available at https://www.arm.gov/data/. The EPA's AQS data is available at https://aqs.epa.gov. Model outputs are available at https://doi.org/10.6084/m9.figshare.21493986.v1.

**Competing interests**

The authors declare no competing interests.

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

**Table 1. Comparison of the SOA schemes used in this study.** Further details and discussions are included in Methods.

| Scheme | ASOA | ISOA | TSOA | | | | | |
|---|---|---|---|---|---|---|---|---|
| Simple | $C_4H_{10}$+OH | 10% yields from isoprene emissions | 10% yields from monoterpene emissions | | | | | |
| CMPX | Same as Simple | Heterogeneous uptake of IEPOX ($\gamma$=0.001) and glyoxal ($\gamma$=0.001)[1] | 4-bin VBS[2] | α for C* (C* in µg/m³) | | | | Yield at 10 µg/m³ |
| | | | | C*=0.1 | C*=1 | C*=10 | C*=100 | |
| | | | MTP+OH/O₃; NO (high-NOₓ pathway) | 0.04 | 0.0095 | 0.09 | 0.015 | 0.09 |
| | | | MTP+OH/O₃; HO₂ (low-NOₓ pathway) | 0.08 | 0.019 | 0.18 | 0.03 | 0.19 |
| | | | MTP+NO₃ | 0 | 0 | 0.321 | 1.083 | 0.26 |
| CMPX_ag[3] | Same as Simple | Same as CMPX | Same as CMPX, with aging $k_{OH}$ = 4×10⁻¹¹ cm³ molec⁻¹ s⁻¹ | | | | | |

[1] $\gamma$ represents uptake coefficients of IEPOX or glyoxal onto aqueous sulfate aerosol.

[2] In the 4-bin VBS, monoterpene (MTP) is oxidized by OH, O₃ or NO₃ to generate 4 semivolatile surrogate products, which can reversibly partition into pre-existing organic aerosol. C* represents saturation concentration of each semivolatile product and determines the partitioning of these products between gas and aerosol phase. The mass-based stoichiometric yield coefficients, α, for each parent hydrocarbon/oxidant system are fit with a VBS using C* of 0.1, 1, 10, and 100 µg/m³ (Pye et al. 2010).

[3] In the aging scheme, at every time step, each semivolatile product except for the lowest volatility bin (C*=0.1 µg m⁻³) is assumed to be further oxidized by OH with a rate constant of $k_{OH}$ = 4×10⁻¹¹ cm³ molec⁻¹ s⁻¹, which reduces its volatility by an order of magnitude.

**Table 2.** Annual mean budget of POA and SOA in all simulations. Results are averaged over 1872-1888 for pre-industrial and 2000-2016 for present-day simulations. SOA includes ASOA (anthropogenic SOA), ISOA (isoprene-SOA), and TSOA (monoterpene-SOA).

| Simulation | Variable | PI | | | | | PD | | | | |
|---|---|---|---|---|---|---|---|---|---|---|---|
| | | Burden (Tg) | Production (Tg/yr) | Wet Deposition (Tg/yr) | Dry Deposition (Tg/yr) | Lifetime (day) | Burden (Tg) | Production (Tg/yr) | Wet Deposition (Tg/yr) | Dry Deposition (Tg/yr) | Lifetime (day) |
| All[1] | POA | 0.58 | 47.0 | 32.4 | 14.6 | 4.5 | 1.00 | 68.1 | 48.9 | 19.2 | 5.4 |
| Simple | ASOA | 0.003 | 0.2 | - | - | - | 0.06 | 3.3 | - | - | - |
| | ISOA | 0.83 | 80.4 | - | - | - | 0.78 | 65.0 | - | - | - |
| | TSOA | 0.15 | 15.6 | - | - | - | 0.15 | 13.4 | - | - | - |
| | Total SOA | 0.98 | 96.2 | 79.3 | 16.9 | 3.7 | 0.99 | 81.7 | 68.0 | 13.7 | 4.4 |
| CMPX | ASOA | 0.003 | 0.2 | - | - | - | 0.06 | 3.3 | - | - | - |
| | ISOA | 0.11 | 10.7 | - | - | - | 0.26 | 22.2 | - | - | - |
| | TSOA | 0.16 | 15.5 | - | - | - | 0.17 | 14.4 | - | - | - |
| | Total SOA | 0.27 | 26.4 | 22.3 | 4.1 | 3.7 | 0.50 | 39.9 | 33.6 | 6.3 | 4.6 |
| CMPX_ag | ASOA | 0.003 | 0.2 | - | - | - | 0.06 | 3.3 | - | - | - |
| | ISOA | 0.11 | 10.8 | - | - | - | 0.27 | 22.0 | - | - | - |
| | TSOA | 0.63 | 40.1 | - | - | - | 0.72 | 44.0 | - | - | - |
| | Total SOA | 0.74 | 51.1 | 43.4 | 7.7 | 5.3 | 1.05 | 69.3 | 58.9 | 10.4 | 5.5 |

[1]For POA budget, the differences between different schemes are negligible.

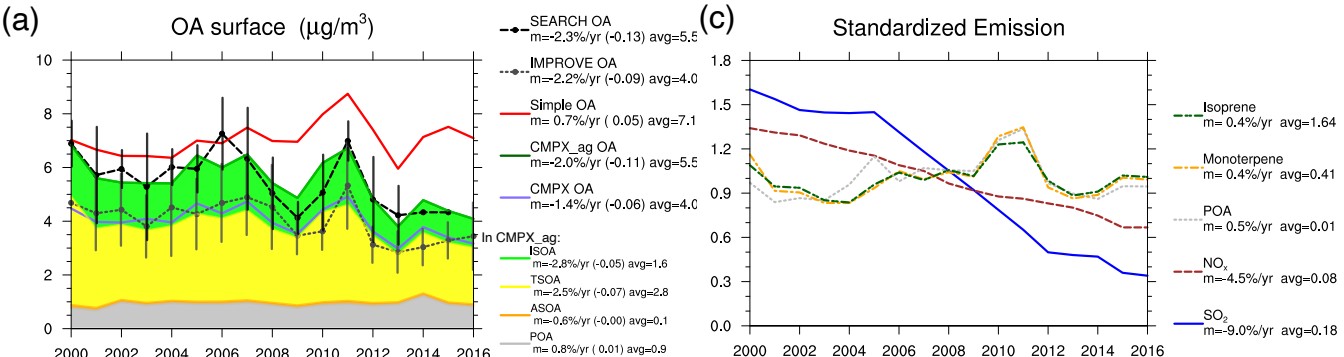

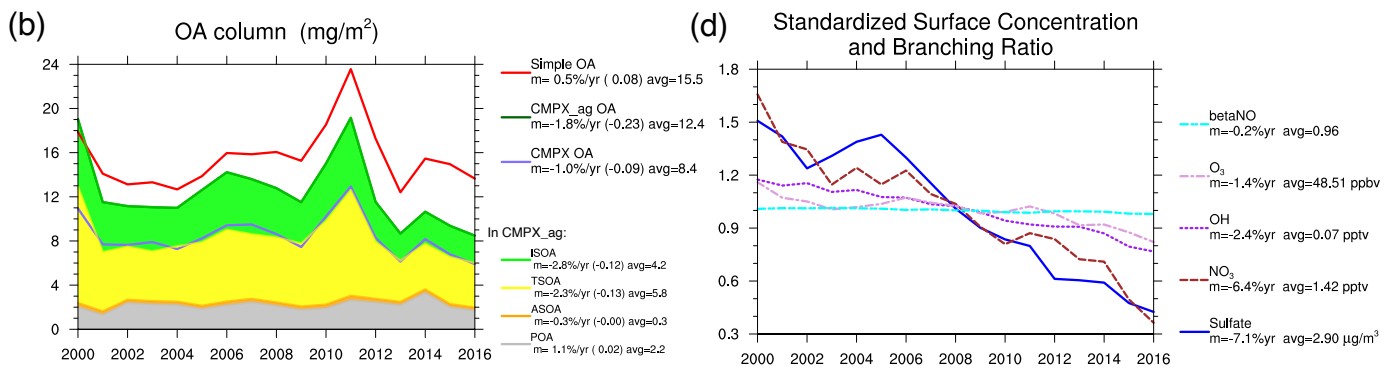

**Figure 1.** Summertime (June-July-August) values averaged in the southeast US (29-37°N, 74-96°W) in 2000-2016. **(a)** Surface concentrations of OA from the two measurement networks, IMPROVE and SEARCH, and the Simple, CMPX and CMPX_ag simulations. **(b)** Column concentrations of OA. In (a) and (b), color shades represent OA components from the CMPX_ag scheme. **(c)** Standardized emissions of isoprene, monoterpenes, POA, $NO_x$ and $SO_2$. **(d)** Standardized surface concentrations of gases $O_3$, OH and $NO_3$, sulfate aerosol, and branching ratio. In (c) and (d), each variable has been divided by its 17-year average for standardization. In attached text, "m" represents 2000-2016 relative trend with units of %/year; numbers in parenthesis in (a) and (b) represent trends with units of $\mu g/m^3$/year or $mg/m^2$/year; "avg" represents the 17-year average with units of $\mu g/m^3$ in (a), $mg/m^2$ in (b), $mg/m^2$/hour in (c) and different units shown in (d). ISOA, TSOA, and ASOA refer to isoprene-, monoterpene-, and anthropogenic-SOA, respectively.

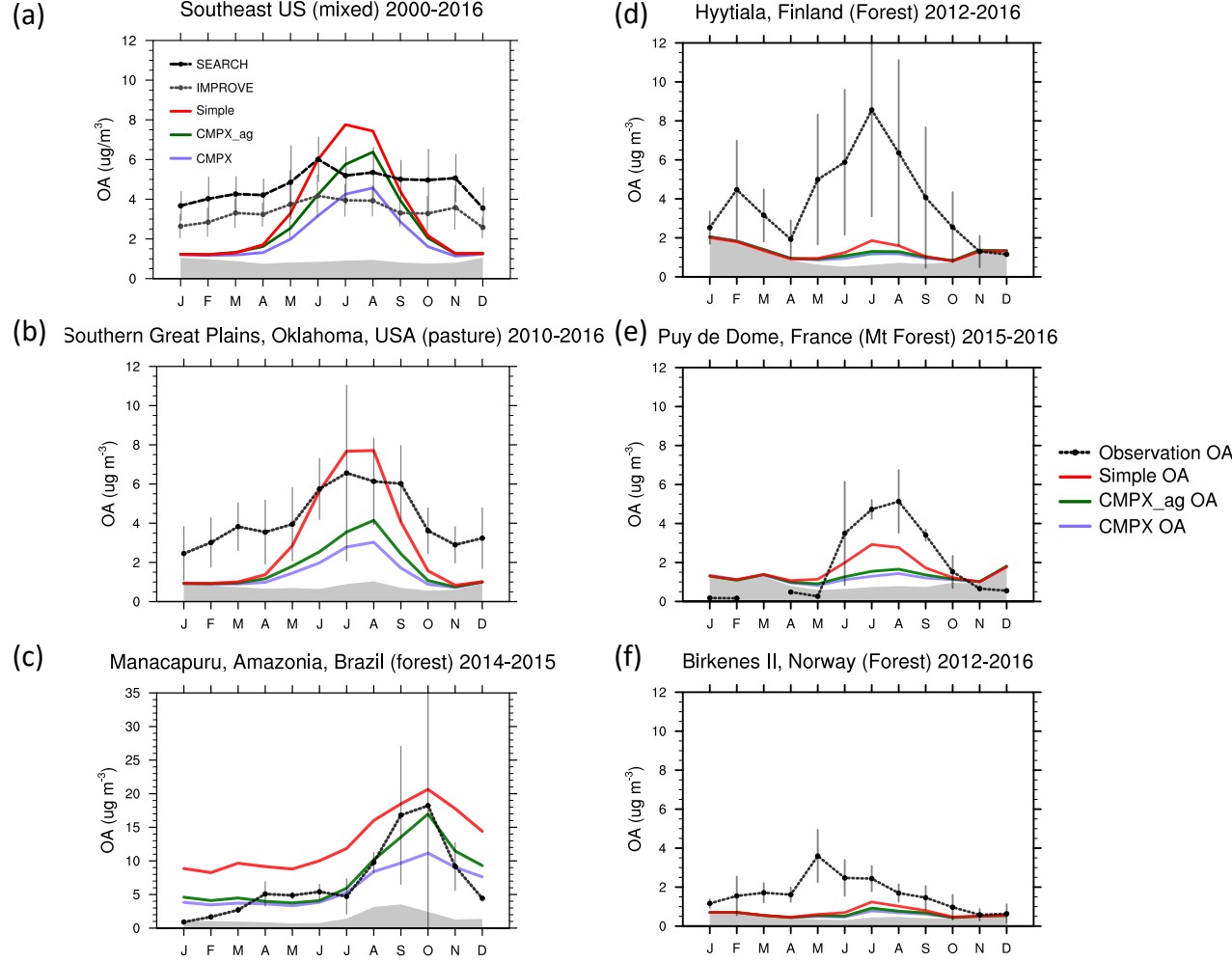

**Figure 2.** Seasonal cycle of surface OA concentration. In (a), black and grey dash lines represent filter measurement of OA from the SEARCH and IMPROVE networks. In other figures, black dash lines represent ACSM measurement of OA from the ARM network in (b)~(c) and from the ACTRIS network in (d)~(f). Grey area represents POA.

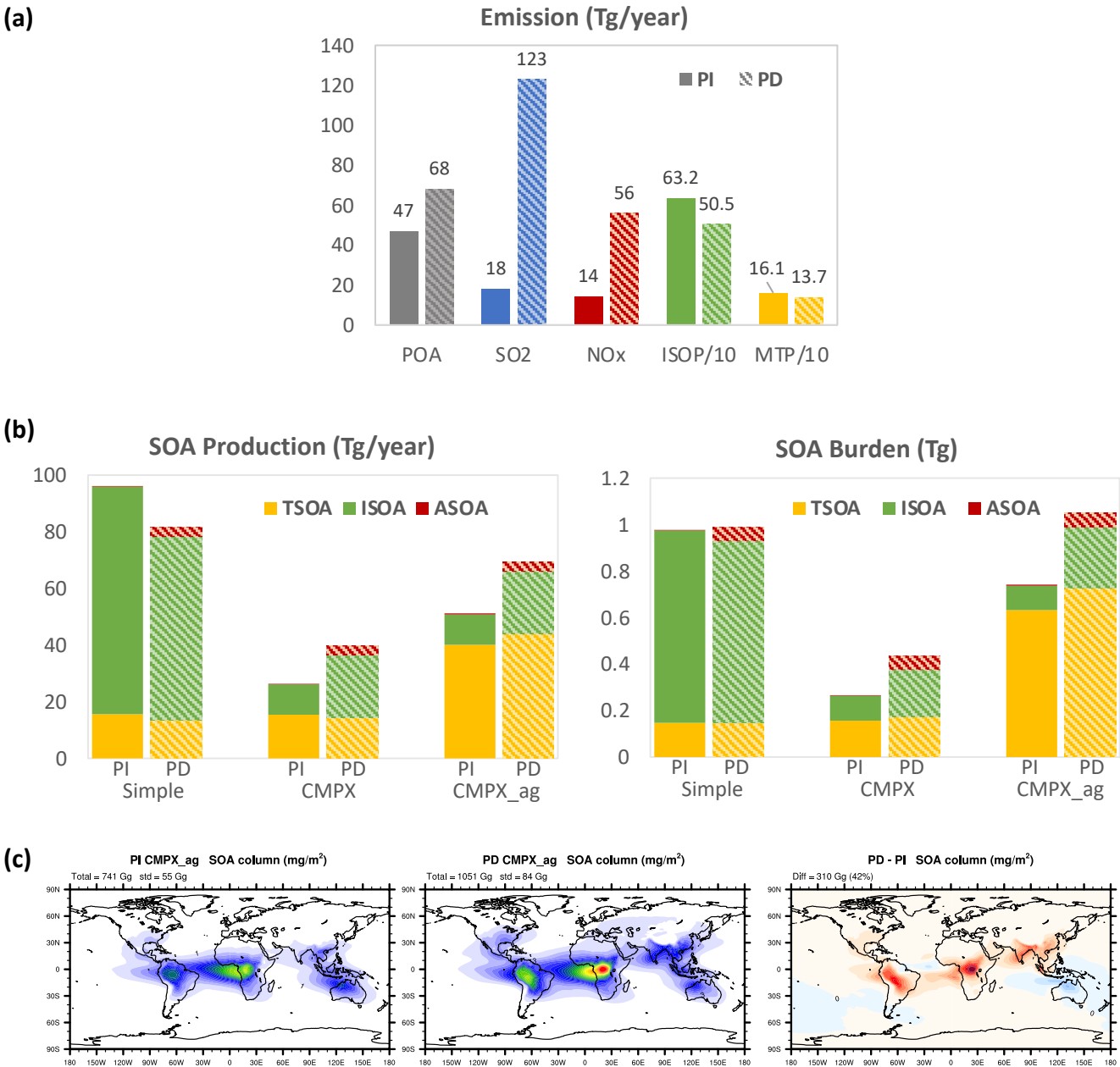

**Figure 3. (a)** Emissions (Tg/year) of POA, $SO_2$, $NO_x$, isoprene (ISOP) and monoterpenes (MTP). ISOP and MTP emissions have been divided by 10. **(b)** Simulated SOA global production (Tg/year) and burden (Tg). **(c)** Simulated SOA column concentration (mg/m$^2$) at PI and PD and their difference in the CMPX_ag scheme.

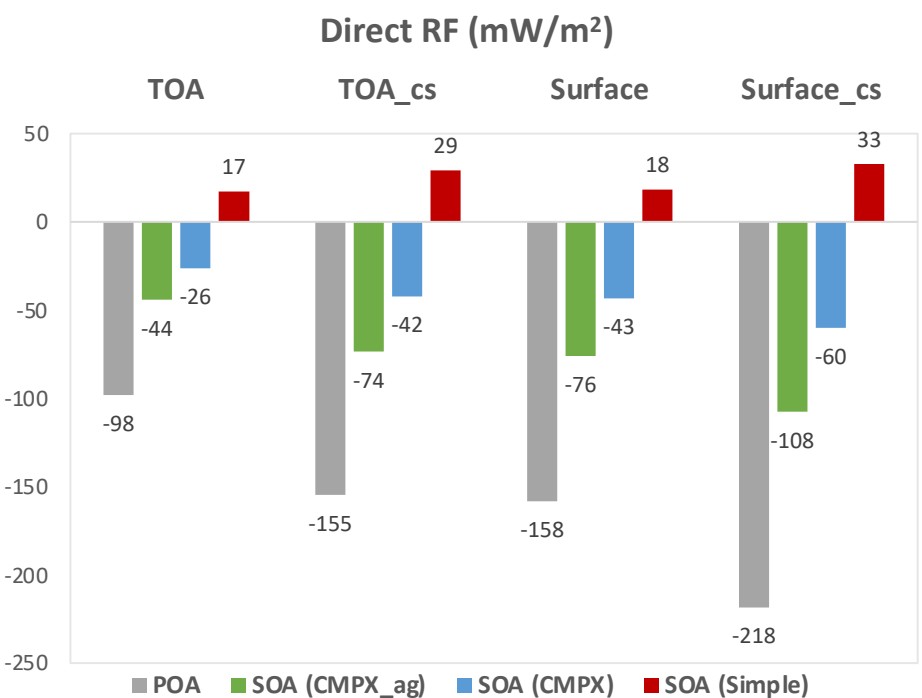

**Figure 4.** Direct radiative forcing (RF, mW/m$^2$) of POA and SOA at top-of-atmosphere (TOA) all-sky, TOA clear-sky (TOA_cs), surface all-sky and surface clear-sky (Surface_cs) conditions. Negative RFs represent cooling effects.