# Peer review of "Anthropogenic amplification of biogenic secondary organic aerosol production 4 5 6 Yiqi Zheng1,2,\*, Larry W. Horowitz3, Raymond Menzel3, David J Paynter3, Vaishali Naik3, Jingyi Li4, Jingqiu Mao1,2,\* 1Geophysic"

_EGUsphere, 2023_

## Author Comment (AC1)

Responses to Reviewer #1

We are grateful to the reviewer for the helpful comments and guidance that led to important improvements of the original manuscript. Our point-by-point responses are listed below. Reviewer's comments are in black font, and authors' responses are in blue. Page and line numbers refer to the manuscript egusphere-2023-372 (the ACPD version).

In this manuscript, Zheng et al. investigate the role of anthropogenic pollutants on the production of biogenic secondary organic aerosols. Using a chemistry-climate model, they perform decadal simulations for present day and pre-industrial periods and compare three schemes that simulate the production of secondary organic aerosols (SOAs) with increasing complexity: (1) Simple, (2) CMPX and (3) CMPX_ag. Over high-source regions of biogenic emissions such as the South-eastern United States and the Amazon, the more advanced scheme, CMPX_ag, outperforms the others and reproduces the observed seasonal variability and trend of summertime organic aerosols. By accounting for different NOx regimes (low vs. high) and for photochemical aging, the CMPX_ag scheme is more sensible to anthropogenic pollution, thus simulating an increased SOA production over present-day compared to the pre-industrial period, although land-cover change has driven a decrease in biogenic emissions. Hence, this study highlights the tight link between SOA production and anthropogenic pollution.

The paper is within the scope of ACP. It examines an important topic such as the sensitivity of biogenic SOA production to anthropogenic pollution, and addresses relevant scientific questions. The paper is well written, the abstract is concise and complete, the introduction is exhaustive and clear, the methods and modeling are well laid out, the literature is thoroughly referenced, and the results are presented in good clear figures. For this reason, I recommend publication after a few minor comments, listed below, have been addressed by the authors.

**Sect. 2, Methods**

Regarding the GFDL AM4.1 model and the modeling of the SOA formation, since the CMPX scheme depends on OH, $O_3$ and $NO_x$ abundance, in my opinion the author should show how well the GFDL AM4.1 model reproduces these gases, or at least insert a sentence/paragraph that summarizes results from previous studies that evaluated the model performance.

To allow the traceability of results, I think it is important to provide details on the spatial resolution and the time-step of the GFDL AM4.1 model. I also suggest to precise the original temporal resolution of observational datasets (IMPROVE, SEARCH, ACTRIS and ARM) and if (and how) these data have been aggregated.

Moreover, I think it is important to explain in the Methods section how relative/percent trends, presented in Sect. 3 (e.g., pag.6, line 18), have been computed. If I correctly understood, I found this information in Fig. S4, in the Supplementary Material ("Changing rate *m* have units of % per year relative to their 2000-2016 averages").

In Page 5 Line 8, we add: "The surrogate TSOA products are implemented in addition to the original gas-phase monoterpene oxidation chemistry in AM4.1 and the implementation does not

doubt count reductions of OH, $O_3$ and $NO_3$. There is little difference in the concentration of these gases between the CMPX and CTRL simulations. The gas-phase chemistry has been validated in Horowitz et al. (2020) and in Figure S3 in which we show summertime $O_3$ and $NO_2$ in SEUS well reproduce their observed decreasing trend."

In Page 4 Line 11, we add: "AM4.1 has 49 vertical levels from surface to 1Pa (~80km). We conduct AM4.1 simulations at a horizontal resolution of 1°×1.25° latitude by longitude."

In Page 5 Line 35, we add: "IMPROVE and SEARCH report daily average organic carbon measurements every 3 days. … and calculate monthly average of organic aerosol (OA) across these sites for each network."

In Page 5 Line 39, we add: "In Section 3.1, we calculate the absolute trend of a variable as the slope of the regression line of the variable's values versus time, and we calculate the relative trend (represented by "m" in Figure 1) as the absolute trend divided by the variable's 2000-2016 average."

In Page 6 Line 2, we add: "We average the original hourly OA measurement to monthly mean data for these sites to compare with modeling results."

**Pag. 4, line 17-18**: In my opinion, I think it is important to specify that LAI values follow an annual cycle (prescribed at the 1992 level, as precised by the authors).

In Page 4 Line 19, we add: "LAI values follow an annual cycle of the year 1992 and PFTs are prescribed at the 1992 level."

**Sect. 3, Results**

**Pag. 6, line 27**: To avoid confusion among readers, figures should be referenced in the order they appear in the text. Here, the authors refer to the Supplementary Figure S5, while in the next paragraph they refer to Fig. S3. For this reason, I suggest the authors to i) revise the order supplementary figures are presented in the Supplementary Material, and ii) comment all the supplementary figures (or remove those that are not commented in the manuscript).

Thanks! We revise and correct all supplement figure numbers accordingly.

**Pag. 7, ll. 21**: I suggest to briefly recall the other vegetated regions that have been selected for evaluation and that have been presented in Sect. 2.3.

In Page 7 Line 21, we add: "… in other vegetated regions in the Amazon, Europe and US (Figure 2)."

**Sect. 4, Summary**

I think that the application of the CMPX_ag scheme could be also interesting for local-regional studies on SOA production. For this reason, although Zheng et al. performed their study at the global scale, I think it could be useful for readers to know the computational cost of including the CMPX_ag scheme in their runs, compared to the cost of using the Simple or the CMPX schemes. This information could be provided in the Methods Section, or it could be commented in the Summary when discussing about perspectives.

As we apply a simplified aging scheme without adding more tracers, the CMPX_ag scheme does not increase the computational cost significantly relative to the CMPX simulation. The difference in their runtimes is almost negligible. In Page 5 Line 16, we add: "Including the aging scheme in CMPX does not increase computational cost notably."

**MINOR COMMENTS**

**Through the whole manuscript:** space is missing before parenthetical citations (e.g., pag.2, line 2: "BVOCs(Guenther et al., 2012))". Please add these spaces.

Corrected.

**Sect. 1, Introduction**

**Pag. 3, ll. 2**: Definition of the acronym CMPX and CMPX_ag are missing. Please define them.

We remove the names of simulations here and save them for the Methods Section. The sentence now is: "We use three schemes (summarized in Table 1 and detailed in Methods) to investigate the AIBS from decadal to centennial time scales."

**Sect. 3, Results**

**Pag. 8, ll. 18**: I think "is" is missing before the adjective "consistent".

The words "which is" are omitted here and should be fine.

**Figures**

**Fig. 1**: In the figure caption, I suggest to recall region boundaries, which are precised in the text (pag. 5, ll. 37). As well, I think it could be useful to recall in the caption the meaning of the different acronyms (ISOA, TSOA, ASOA)

We add the region boundaries and the definition of ISOA, TSOA and ASOA in the figure caption.

---

## Author Comment (AC2)

Responses to Reviewer #2

We are grateful to the reviewer for the helpful comments and guidance that led to important improvements of the original manuscript. Our point-by-point responses are listed below. Reviewer's comments are in black font, and authors' responses are in blue. Page and line numbers refer to the manuscript egusphere-2023-372 (the ACPD version).

**General Comments**

The authors tackle an important and underdeveloped scientific question (the relative influence of anthropogenic emissions on biogenic SOA) and the analysis has clearly involved a meaningful and thoughtful allocation of time and resources (model runs + observational comparisons). The authors highlight a few key results around the importance of appropriately modeling the anthropogenic impact on biogenic SOA formation (AIBS) in order to (1) better explain the trends in SOA over the past 2 decades and (2) provide insight into the anthropogenic influence on SOA (and associated climate impacts) relative to pre-industrial atmospheres. The study also highlights an important result regarding the relative influence SO2 vs NOx (and ISOA vs TSOA). The authors also use observational constraints to validate their models.

While the study provides valuable insight into this important domain, the SOA mechanisms themselves could be better validated, more varied and more detailed. Given that the central goal of the manuscript is to highlight the importance of AIBS, there could be value to exploring other model SOA mechanisms that explicitly vary in their sensitivity to NOx and SO2. A more comprehensive mechanistic comparison of such schemes would provide greater certainty to some of the central results in this paper (such as the outsized importance of NOx vs SO2 in explaining recent SOA trends). This paper could also be meaningfully strengthened if the authors deepened their analysis of the ISOA & TSOA pathways via a discussion and literature review of the current state of the science in modeling the anthropogenic impact on both species and a more detailed focus on the underlying uncertainties in their approach to modeling each of the three schemes (for instance, the simple mechanism which assumes a 10% yield for ISOA is likely overestimating this parameter based on other literature in this space. While the authors do allude to this, a more detailed discussion and yield sensitivity analysis would strengthen the model comparison that ties together their core argument).

Overall, the authors tackle an important scientific question and the resulting analysis fits well within the scope of ACP. I would recommend the publication of this manuscript following the revisions outlined below that would strengthen its contribution to the domain.

We are thankful to the reviewer's comments, and we totally agree that explicit, semi-explicit or semi-empirical SOA mechanisms with various complexity show different relative importance of NOx and SO2, and there is no consensus yet. Our mechanism only provides one possible explanation. Below we discuss these mechanisms and uncertainties in the following response to "OA Mechanism Selection and Modification" and "Mechanistic learnings".

**Specific Comments**

**• OA Mechanism Selection and Modification**

The paper would benefit from a more detailed overview of the different SOA schemes and why they were chosen / modified in the manner they were (as opposed to other mechanistic changes that might also have nudged them in the right direction given the authors' hypothesis). A more detailed overview / discussion of the impact that implementing other mechanisms (such as an explicit auto-oxidation mechanism, alternate IEPOX mechanisms, etc.) might have had on the CMPX_ag simulation would also be helpful.

In Page 4 Line 40, we add: "This 10% yield in the Simple scheme is consistent with previous model versions GFDL AM3 and AM4.0 and within the range of estimates suggested by other studies. For example, a chemical transport model GEOS-Chem assumed a 3% yield for isoprene and a 10% yield of monoterpene emissions (Pye et al., 2010; Pai et al., 2020). However, a study using a more complex scheme suggested a SOA yield from isoprene of 13% per carbon (Bates and Jacob, 2019)."

In Page 5 Line 19, we previously included the discussion on the choice of ISOA mechanism relative to previous schemes: "The uptake coefficients for IEPOX and glyoxal are set to 0.001, different than previous studies using higher or acidity-dependent uptake coefficients (Marais et al., 2016; Lin et al., 2014a). This is supported by the OA month-to-month variability (MMV) in summer and its decadal trend over the southeast US, as a previous model with acidity-dependent uptake coefficients shows too high of MMV and too much OA in the early 2000s (Zheng et al., 2020). The uptake rate coefficients can be even lower due to the effect of aerosol-phase state (Zhang et al., 2018b). To avoid uncertainties associated with aerosol acidity, relative humidity, and coating effect, we here apply uptake coefficient of 0.001 for both IEPOX and glyoxal. This leads to good agreement between our model and observation in the SEUS on both OA magnitude and summertime MMV (Figure1, S1, S2 and S13)."

In Page 5 Line 9, we add: "Such semi-empirical partitioning-based VBS schemes have been widely used in chemistry-climate models and Earth system models (e.g. Zheng et al., 2015; Tilmes et al., 2019). Recent research show that these schemes may underestimate SOA formation without considering further aging processes, such as oligomerization in the organic phase and aqueous-phase reactions (Hu et al., 2013; Yu et al., 2021; Oak et al., 2022 and references therein). One major recent identified explicit mechanism is the formation of monoterpene-derived highly oxygenated molecules (HOMs) through the autooxidation of peroxy radicals (Crounse et al., 2013; Ehn et al., 2014; Pye et al., 2019). Mechanistic schemes of monoterpene-derived SOA have been developed with varying complexity at a cost of more tracers and reactions (Pye et al., 2019; Berkemeier et al., 2020; Pullinen et al., 2020; Yu et al., 2021), which may not be mature for a global climate model as part of an Earth system model considering large uncertainties associated with multi-phase processes and increased computational cost. In this study, in addition to the semi-empirical VBS scheme, we implement a simplified photochemical aging parameterization to the semivolatile oxidation products of terpenes in the CMPX scheme (CMPX_ag). … This simplified aging scheme does not explicitly represent up-to-date knowledge of SOA chemistry but similarly increases the SOA burden as well as the sensitivity of SOA to $NO_x$, improving the model underestimate of SOA by the VBS scheme."

• **Model Specifics**

A more detailed overview and discussion of the general model simulation such as resolution, time-step, loss processes, pre-industrial emissions etc. would be useful in reproducing these results. More details / uncertainties regarding the modelled emissions and atmospheric fates of the anthropogenic gas-phase species that influence the SOA production ($NO_x$, $SO_2$, etc.) would be similarly useful.

In Page 4 Line 11, we add: "AM4.1 has 49 vertical levels from surface to 1Pa (~80km). We conduct AM4.1 simulations at a horizontal resolution of 1°×1.25° latitude by longitude and a main dynamical atmosphere time step of 30 minutes. Annual varying historical anthropogenic emissions from pre-industrial era to present day (1849 to 2016) are from the Community Emissions Data System (CEDS) (Hoesly et al., 2018) and the data set of van Marle et al. (2017), which are developed in support of the Coupled Model Intercomparison Project Phase 6 (CMIP6)."

In Page 4 Line 19, we add: "The gas-phase and aerosol chemistry is detailed in Horowitz et al. (2020), in which heterogenous reactive uptake of $HO_2$, $HO_2$, $NO_2$, $N_2O_5$, $NO_3$, $SO_2$, IEPOX and glyoxal onto aerosol surfaces are included. Dry and wet deposition of gases are described in Paulot et al. (2016). More details could be found in Horowitz et al. (2020) and Dunne et al. (2020)."

• **Observations**

Similarly, the manuscript would benefit from a more detailed overview of how the observations were sampled, averaged and manipulated. More detailed statistics / uncertainties on the model-observation comparison would also be useful. Lastly, a detailed discussion on the kind of observations that might prove out the core thesis of this work (e.g. the relative importance of $NO_x$ vs $SO_2$) would help future research build on this study.

In Page 5 Line 32, we add: "For model evaluation we use long-term measurements of organic aerosol (OA) or organic carbon (OC). We do not use explicit SOA tracers in this study because it is not suitable to use short-term observations to validate simulated results from a chemistry-climate model like AM4.1, in which meteorology is not offline specified by reanalysis data, but is free running in the dynamic core. However, long-term (covering periods of at least months to years) measurement of explicit SOA species is not available."

In Page 5 Line 35, we add: "IMPROVE and SEARCH report daily average organic carbon measurements every 3 days. … and calculate monthly average of organic aerosol (OA) across these sites for each network." In Page 6 Line 2, we add: "We average the original hourly OA measurement to monthly mean data for these sites to compare with modeling results."

In Page 6 Line 20, we modify the text as follows: "From 2000 to 2016, the measured summer OA declines by -0.13 $\mu g/m^3$/year from SEARCH and by -0.09 $\mu g/m^3$/year from IMPROVE, both at a reduction rate of -2.3%/year (Figure 1a). This decreasing trend is well reproduced by the CMPX_ag simulation with a decrease of -0.11 $\mu g/m^3$ (-2.0%) per year, and a smaller decrease of -

0.06 μg/m$^3$ (-1.4%) per year with the CMPX scheme. Considering the varying reduction trends among different sites (Figure S4), both the CMPX and CMPX_ag schemes well reproduce the SEUS OA trend in general."

We move Figure S6 to Figure 2 to show the comparison with ACSM measurements in other vegetated regions. In Page 7 Line 22, we add: "In the Amazon region, the CMPX_ag scheme successfully reproduces the high surface OA concentration from August to November and low OA in other months. The Simple scheme greatly overestimates surface OA in all seasons because of its high SOA yield (10%) from isoprene emissions. The CMPX scheme well reproduces the low OA concentrations from January to July but only predicts half of observed OA in months with high OA concentrations." In Page 7 Line 30, we add: "The good performance of the CMPX_ag scheme in the Amazon, better than the CMPX scheme, gives us confidence that the traditional VBS in the CMPX scheme may underestimate the contribution of TSOA and its sensitivity to NO$_x$."

Please also see the response below that summarizes the observation-model comparison and interpretation.

• **Mechanistic Learnings**

Since the paper makes a point to stress the mechanistic relevance of anthropogenic pathways for SOA, it could be meaningfully strengthened by a more detailed mechanistic analysis / comparison of these pathways (either via more model runs that isolate their relative importance or via a detailed discussion and literature review that hypothesizes the relative importance of other mechanisms not considered in this analysis). For instance - Do the authors recommend the CMPX_ag as the SOA scheme of choice? What are the benefits / downsides of using this scheme relative to other complex schemes in different community models? In the authors opinion, does this analysis close much of the gap in our understanding of AIBS or are there still large undeveloped mechanisms that need to be explored? If NOx is more influential than SO2 in describing recent SOA trends, are there implications from this study on how atmospheric aerosol models should treat the emissions / fate of NOx / SO2 going forward?

In Page 9 Line 16, we modify the summary text as follows: "Constrained by observations in SEUS, we show that the summertime OA decreasing trend is likely driven by reduction in both NO$_x$ and SO$_2$ emissions, through TSOA and ISOA. First, in a previous study (Zheng et al., 2020) we prove that the scheme of acidity-catalyzed aqueous ISOA formation (Marais et al., 2016) strongly overestimates summertime month-to-month variability of surface OA, therefore in this study we use fixed uptake coefficients for isoprene oxidation products to avoid uncertainties associated with acidity, relative humidity, and coating effect. Second, both the CMPX and CMPX_ag schemes reproduce the observed OA magnitude and the decadal trend in SEUS, in which SO$_2$ alone cannot explain this trend. The CMPX_ag scheme shows a faster OA decrease and better agrees with long-term filter measurement, which is largely driven by NO$_x$ (60%). Third, the CMPX_ag scheme successfully reproduces the observed OA magnitude and seasonal cycle in Amazon, outperforming the CMPX and Simple schemes. Our results point to the importance role of NO$_x$ on modulating biogenic SOA, in line with recent understanding on autooxidation (Crounse et al., 2013; Ehn et al., 2014; Pye et al., 2019), although further studies are warranted. For example, the CMPX_ag scheme with a simplified aging parameterization

[revised manuscript text omitted]